# In Situ Construction of Bronze/Anatase TiO_2_ Homogeneous Heterojunctions and Their Photocatalytic Performances

**DOI:** 10.3390/nano12071122

**Published:** 2022-03-29

**Authors:** Yong Li, Ming-Qing Zhang, Yan-Fang Liu, Ya-Xun Sun, Qing-Hua Zhao, Tian-Lu Chen, Yuan-Fu Chen, Shi-Feng Wang

**Affiliations:** 1Innovation Center of Materials for Energy and Environment Technologies, College of Science, Tibet University, Lhasa 850000, China; xzuliyong@utibet.edu.cn (Y.L.); mingqing@utibet.edu.cn (M.-Q.Z.); liuyanfang@utibet.edu.cn (Y.-F.L.); sunyaxun@utibet.edu.cn (Y.-X.S.); zhaoqinghua@utibet.edu.cn (Q.-H.Z.); 2Institute of Oxygen Supply, Center of Tibetan Studies (Everest Research Institute), Tibet University, Lhasa 850000, China; 3Key Laboratory of Cosmic Rays (Tibet University), Ministry of Education, Lhasa 850000, China; 4School of Electronic Science and Engineering, and State Key Laboratory of Electronic Thin Films and Integrated Devices, University of Electronic Science and Technology of China, Chengdu 610054, China

**Keywords:** homogeneous heterojunctions, photocatalyst, bronze TiO_2_, anatase TiO_2_, photocatalytic degradation

## Abstract

Photocatalytic degradation is one of the most promising emerging technologies for environmental pollution control. However, the preparation of efficient, low-cost photocatalysts still faces many challenges. TiO_2_ is a widely available and inexpensive photocatalyst material, but improving its catalytic degradation performance has posed a significant challenge due to its shortcomings, such as the easy recombination of its photogenerated electron–hole pairs and its difficulty in absorbing visible light. The construction of homogeneous heterojunctions is an effective means to enhance the photocatalytic performances of photocatalysts. In this study, a TiO_2_(B)/TiO_2_(A) homogeneous heterojunction composite photocatalyst (with B and A denoting bronze and anatase phases, respectively) was successfully constructed in situ. Although the construction of homogeneous heterojunctions did not improve the light absorption performance of the material, its photocatalytic degradation performance was substantially enhanced. This was due to the suppression of the recombination of photogenerated electron–hole pairs and the enhancement of the carrier mobility. The photocatalytic ability of the TiO_2_(B)/TiO_2_(A) homogeneous heterojunction composite photocatalyst was up to three times higher than that of raw TiO_2_ (pure anatase TiO_2_).

## 1. Introduction

Environmental pollution is one of the biggest crises facing humans, and many environmental treatment technologies have been developed to control pollution [1,2]. Photocatalytic degradation has become one of the most promising emerging environmental pollution control technologies due to its mild reaction conditions, lack of secondary pollution generation, and environmental friendliness [3,4,5,6]. The key to the practical large-scale application of photocatalysis is the preparation of efficient, low-cost photocatalysts. However, most currently available photocatalysts fail to achieve a balance between a high efficiency and a low cost [7,8]. The modification of inexpensive, widely available photocatalysts to enhance their photocatalytic performances has become an important research direction for photocatalytic degradation technology [9,10,11]. TiO_2_ is a non-toxic, non-hazardous, and low-cost photocatalyst with stable performance, but it exhibits poor photocatalytic performance, which must be enhanced to meet the needs of practical applications [12,13,14,15,16]. The low performance of TiO_2_ photocatalysts is attributed to the easy recombination of photogenerated electron–hole pairs and the difficulty of visible light absorption. The construction of a homogeneous heterojunction is an effective method for suppressing the recombination of photogenerated electron–hole pairs and enhancing the carrier mobility, thereby improving the photocatalytic performance of photocatalysts [17,18,19,20,21,22]. As a rare phase in TiO_2_, TiO_2_(B) (bronze TiO_2_) has a good electrical conductivity, a loose, porous structure, and it can be constructed with TiO_2_ in the anatase phase (denoted as TiO_2_(A)), to form a homogeneous heterojunction crystal lattice with very small mismatches. As a result, the TiO_2_(B)/TiO_2_(A) homogeneous heterojunction will exhibit good photocatalytic degradation performance [23,24,25,26]. In this study, TiO_2_(B)/TiO_2_(A) homogeneous heterojunctions were constructed in situ by a simple hydrothermal method, utilizing TiO_2_(A) as the raw material with ion exchange and calcination. The prepared TiO_2_(B)/TiO_2_(A) homogeneous heterojunction material exhibited an excellent photocatalytic degradation performance, which was up to three times higher than that of raw TiO_2_. At present, the research on TiO_2_ homogeneous heterojunctions mostly focuses on anatase/rutile TiO_2_, while much less research focuses on bronze/anatase TiO_2_. This study will provide new ideas and beneficial attempts for the research of TiO_2_ homogeneous heterojunctions.

## 2. Materials and Methods

### 2.1. Materials

The rhodamine B (RhB, C_28_H_31_CIN_2_O_3_, Purity: 95%) was purchased from china Beijing Solarbio Technology, and both the sodium hydroxide (NaOH, Purity: 96%) and nano titanium dioxide (TiO_2_, Purity: 99.8%) were purchased from china Shanghai Aladdin Industrial Company. All reagents used in the experiments were of analytical grade and were used without further purification.

### 2.2. Preparation of Samples

A 100 mL 10 M NaOH aqueous solution was prepared, and 1 g raw TiO_2_ was added. The mixture was stirred well, poured into a 150 mL hydrothermal reactor, heated to 220 °C in a blast drying oven, and held for 8 h. The reaction product was taken out and rinsed with a large amount of deionized water to near neutrality, after which it was dried at 60 °C for 10 h to obtain an intermediate product, which was labeled DNaT-220. The DNaT-220 was then soaked in 1000 mL 0.1 M dilute hydrochloric acid for 72 h, rinsed with a large amount of deionized water to near neutrality, dried at 60 °C for 10 h, and finally calcined in a muffle furnace at 500 °C for 2 h with a heating rate of 5 °C/min to obtain the final product, which was labeled DT-220. The intermediate products DNaT-200, DNaT-190, DNaT-180, and DNaT-160 as well as the final products DT-200, DT-190, DT-180, and DT-160 were prepared in the same way by changing only the temperature of the hydrothermal reaction. The number after the hyphen indicates the reaction temperature in °C.

### 2.3. Analysis and Testing

The phases of the samples were tested and analyzed using X-ray powder diffraction (XRD, D8 Advance, Bruker, Bremen, Germany) operated with a Cu target, wide-angle diffraction, a scanning range of 10–70°, a scanning rate of 10°/min, a test wavelength of 1.5406 Å, a tube voltage of 40 kV, and a tube current of 40 mA. The morphologies and structures of the samples were observed and analyzed using a field-emission scanning electron microscope (FE-SEM, Gemini SEM 300, Zeiss, Oberkochen, Germany) operated at a working voltage of 3 kV. The morphologies and structures of the samples were also observed and analyzed using a field-emission transmission electron microscope (Talos F200X, FEI, Hillsboro, OR, USA) operated at a working voltage of 200 kV, with a maximum magnification of 1.1 million times and a limit resolution of 0.12 nm. The photocatalytic degradation performance for RhB and the ultraviolet–visible (UV-vis) absorption spectra of the samples were tested by UV-vis spectrophotometry (UV-3600, Shimadzu, Japan) in the wavelength range of 200–2500 nm. The photocurrents and impedances of the samples were tested using an electrochemical workstation (CHI-760E, Chinstruments, Shanghai, China).

To test the photocatalytic degradation performance, 2 L of 20 mg/L RhB solution was prepared. In each experiment, 100 mL of RhB solution was added to 20 mg of the sample, and the mixture was sonicated for 30 min. Then, the mixture was placed under the light source of a solar simulator (Solar-500Q, Newbit, Beijing, China) with an intensity of 600 W/m^2^ (at the liquid surface), and the photocatalytic degradation experiment was performed for 60 min. Every 10 min, 5 mL of liquid was taken from the mixture and placed in a centrifuge for centrifugation at 10,000 rpm for 10 min. After this, the supernatant was removed, and the absorbance (A_n_) was measured by UV-vis spectrophotometry. The solution concentration (C_n_) was proportional to A_n_, and thus, C_n_/C_0_ = A_n_/A_0_, where A_0_ is the absorbance of the 20 mg/L RhB solution.

## 3. Results and Discussion

Figure 1a shows the XRD patterns of the raw TiO_2_ and the intermediate products DNaT-220, DNaT-200, DNaT-190, DNaT-180, and DNaT-160. The raw TiO_2_ diffraction peaks matched well with the standard PDF card JCPDS#21-1272, which corresponds to the most common anatase TiO_2_ phase. The XRD patterns of the DNaT-220 and DNaT-200 had identical peak positions and shapes, coinciding with the PDF card JCPDS#31-1329, which indicated that both samples contained the same Na_2_Ti_3_O_7_ phase. The peaks of the samples with lower hydrothermal reaction temperatures, DNaT-190, DNaT-180, and DNaT-160, changed significantly, which mainly manifested as increases in the half-peak widths of the spectral lines. All the samples contained Na_2_Ti_3_O_7_. However, the positions of the strongest and the second strongest peaks (corresponding to the TiO_2_ in the anatase phase) were close to those of the peaks of Na_2_Ti_3_O_7_, which were relatively strong. Therefore, the peaks of the anatase TiO_2_ phase (corresponding to the incompletely reacted raw TiO_2_) were not clearly observed. 

Figure 1b shows the XRD patterns of the raw TiO_2_ and the final products—DT-220, DT-200, DT-190, DT-180, and DT-160. The peak positions and shapes of the DT-220 and DT-200 spectra were identical, and their standard PDF card number was JCPDS#35-0088, which corresponded to the bronze TiO_2_ phase. When the hydrothermal reaction temperature dropped to 190 °C, the diffraction peaks of the bronze TiO_2_ phase (corresponding to the product DT-190) were significantly weakened, whereas significant diffraction peaks of the anatase TiO_2_ phase were observed. As the hydrothermal reaction temperature continued to be reduced, only the diffraction peaks completely consistent with those of raw TiO_2_ were observed in the XRD patterns of the products DT-180 and DT-160. The TiO_2_·xH_2_O produced by the generated Na_2_Ti_3_O_7_ via ion exchange was converted to TiO_2_(B) after calcination at 500 °C. 

The above phase analysis revealed that the entire phase-transformation process under the present experimental conditions was as follows: At hydrothermal reaction temperatures of 220 °C and 200 °C, the raw TiO_2_ was completely converted by hydrothermal reactions to Na_2_Ti_3_O_7_, which then led entirely to the product TiO_2_(B) after ion exchange and high-temperature calcination. At a hydrothermal reaction temperature of 190 °C, the raw TiO_2_ was partly converted via a hydrothermal reaction to Na_2_Ti_3_O_7_, and then ion exchange and high-temperature calcination led partly to the product TiO_2_(B) and partly to the incompletely reacted anatase TiO_2_ phase. With a further decrease in the hydrothermal reaction temperature, the anatase phase TiO_2_ was converted to Na_2_Ti_3_O_7_ at a slower rate, the proportion of Na_2_Ti_3_O_7_ converted by the reaction further decreased, and the amount of the remaining unreacted anatase TiO_2_ phase further increased accordingly. Through controlled hydrothermal reactions, the structure of the raw TiO_2_ could be easily modulated between anatase and bronze phases, providing a method for homogeneous heterojunction formation in situ, as well as electrical property adjustment.

Figure 2a–d shows the scanning electron microscopy (SEM) images of the DNaT-200, DNaT-190, DNaT-180, and DNaT-160, respectively. Figure 2e–h shows the SEM images of the DT-200, DT-190, DT-180, and DT-160, respectively. The comparison of the two sets of SEM images shows that the morphologies of the final products (DT-200 through DT-160) and the intermediate products (DNaT-200 through DNaT-160) remained the same and unchanged. Combined with the XRD analysis results, we concluded that the band-like material in the two sets of SEM images was Na_2_Ti_3_O_7_ (Figure 2a,b), which retained this morphology after ion exchange and calcination (Figure 2e,f) but was converted to TiO_2_(B). These band-like Na_2_Ti_3_O_7_ and TiO_2_(B) were also observed in samples DNaT-220 and DT-220, respectively. When the hydrothermal reaction temperature dropped to 190 °C, a three-dimensional network started to appear in the DNaT-190 and DT-190 samples and many fine particles were dispersed on the network; these were TiO_2_(A) crystals that had not been converted. As the hydrothermal reaction temperature was further reduced, the conversion rate of the raw TiO_2_ to Na_2_Ti_3_O_7_ decreased. When the hydrothermal reaction temperature decreased to 180 °C, band-like Na_2_Ti_3_O_7_ could no longer be generated by the reaction. Morphologically, the product consisted of Na_2_Ti_3_O_7_ with a network structure and granular TiO_2_(A) dispersed in the network structure that had not completely reacted, as shown in Figure 2c,d,g,h, respectively.

As shown in Figure 3, further transmission electron microscopy (TEM) observation of the DT-190 and DT-180 revealed that DT-190 consisted of band-like TiO_2_(B) and short, rod-like inclusions. Based on the calibration of the interplanar spacing obtained by high resolution TEM (HRTEM), the DT-190 was composed of TiO_2_(B) and TiO_2_(A), and homogeneous heterojunctions constructed by TiO_2_(B) and TiO_2_(A) were observed. The DT-180 consisted entirely of short, rod-like inclusion particles, and homogeneous heterojunctions constructed by TiO_2_(B) and TiO_2_(A) were observed.

Figure 4a shows the photocatalytic degradation curves of the raw TiO_2_, DT-200, DT-190, DT-180, and DT-160, respectively (Each sample was tested three times and the average value was taken). The photocatalytic degradation performances of the two-phase composite products, DT-190, DT-180, and DT-160, composed of TiO_2_(B) and TiO_2_(A) were significantly higher than those of the raw TiO_2_ and pure TiO_2_(B). The DT-180 significantly outperformed the DT-190 and DT-160, and the photocatalytic degradation performance of the DT-180 heterojunction composite was nearly three times higher than that of raw TiO_2_. In addition, we can see that the samples had a small amount of adsorption on RhB during 30 min of ultrasound. Figure 4b quantitatively compares the catalytic efficiencies of the above samples, according to the kinetic calculation results. In terms of the Langmuir–Hinshelwood model, degradation can be considered as a first-order reaction at low RhB concentration. Therefore, the fitting calculation of degrading kinetics can be simplified to be linear, and this reaction satisfies the apparent first-order reaction rate equation: ln(C/C_0_) = −*k*ˑt, where *k* is the apparent rate constant of the first-order reaction, and ln(C/C_0_) is a function of irradiation time *t*. The calculated *k* values of no catalyst, Raw TiO_2_, DT-200, DT-190, DT-180 and DT-160 are 0 min^−1^, 0.004 min^−1^, 0.006 min^−1^, 0.013 min^−1^, 0.020 min^−1^ and 0.014 min^−1^, respectively. These results suggest that the ratio of TiO_2_(B) to TiO_2_(A) in the product was related to the photocatalytic performance of the TiO_2_(B)/TiO_2_(A) homogeneous heterojunction composite photocatalyst, and that the composite exhibited the best photocatalytic performance when the optimal ratio was approached. In addition, the recyclability of DT-180 in RhB degradation reaction under UV-Vis irradiation was investigated, as shown in Figure 4c. After three successive cycles, the DT-180 sample retained nearly consistent photocatalytic efficiency, indicating that the TiO_2_(B)/TiO_2_(A) homogeneous heterojunction composite is stable during the degrading process.

Figure 5a shows the UV-vis absorption spectra of the raw TiO_2_, DT-200, and DT-180. Compared with those of raw TiO_2_ and pure TiO_2_(B), the light absorption performance of the TiO_2_(B)/TiO_2_(A) homogeneous heterojunction composite photocatalyst in the UV region was not improved, but rather decreased due to the successful construction of homogeneous heterojunctions. The light absorption performance in the visible region was not significantly improved, with only a very slight red-shift. Figure 5b shows the Tauc plot of the raw TiO_2_, DT-200, and DT-180. The band gap energy of the semiconductors can be obtained from the *x*-axis intercept of the extended line. From the graph, the band gap energy values for the above samples were determined to be 3.33 ev, 3.18 ev, and 3.24 ev, respectively, which suggests that the band gap energy of TiO_2_(A) was reduced after introducing TiO_2_(B) to TiO_2_(A) to form the homogeneous heterojunction composite.

To investigate the origins of the improved photocatalytic performance, the light responses and electrical properties were measured. Figure 6a shows the transient photocurrent responses of the raw TiO_2_, DT-200 (representative of TiO_2_ in the bronze phase), and DT-180. The photocurrent of DT-200 was higher than that of the raw TiO_2_ in the anatase phase by a factor of approximately 1.5, and that of DT-180 was significantly higher than that of DT-200, by a factor of more than 1.5. Usually, a higher photocurrent indicates a weaker recombination of photogenerated electrons and holes. Thus, the recombination of photogenerated electron–hole pairs was significantly suppressed when the structure of TiO_2_ was transformed from the anatase phase into the bronze phase. This suggests that TiO_2_(B) exhibited a slower recombination rate of photogenerated carriers than TiO_2_(A). In addition, the formation of TiO_2_(B)/TiO_2_(A) homogeneous heterojunctions further boosted the photogenerated charge separation because of the small built-in potential [25,27], leading to the substantially increased photocurrent shown in Figure 6a. Figure 6b shows the electrochemical impedance plots of the raw TiO_2_, DT-200, and DT-180. Generally, a larger arc radius indicates a higher impedance of the sample. Therefore, it was concluded that the impedance of DT-200 was lower than that of the raw TiO_2_, while that of DT-180 was much lower than those of both the raw TiO_2_ and DT-200. This indicated that the conductivity and carrier mobility were improved by the transformation of TiO_2_(A) to TiO_2_(B) and were further enhanced by the formation of TiO_2_(B)/TiO_2_(A) homogeneous heterojunctions. Overall, the construction of TiO_2_(B)/TiO_2_(A) homogeneous heterojunctions not only benefitted the charge separation in the photocatalytic process, but also improved the electrical characteristics. The synergistic effects generated excellent photocatalytic activities, showing great potential for low-cost and high-efficiency photodegradation catalysts in environmental applications.

## 4. Conclusions

In this study, a novel photocatalyst consisting of TiO_2_(B)/TiO_2_(A) homogeneous heterojunctions was successfully prepared through facile hydrothermal processes. The structural characterizations revealed that a transformation occurred in the structure from the anatase phase to the bronze crystal phase, during which a series of TiO_2_(B)/TiO_2_(A) homogeneous heterojunctions formed. No significant improvement in the light absorption was observed, except for a very small red-shift in the absorption spectra. However, a substantial increase in photocurrent was detected for TiO_2_(B) in the transient light response measurement compared to that of TiO_2_(A). Additionally, further growth of the photocurrent for the homogeneous heterojunction was clearly observed, which was likely due to the small built-in field at the heterojunction interface. Moreover, the electrochemical impedance test revealed that the TiO_2_(B)/TiO_2_(A) homogeneous heterojunction possessed a substantially higher conductivity than that of TiO_2_(A), which also benefited the photocatalytic activities. The results of this study also indicated that the photocatalytic degradation performance of the TiO_2_(B)/TiO_2_(A) homogeneous heterojunction composite photocatalyst can be further enhanced by improving its light absorption capacity. This work provides a new structural design concept and a simple preparation method for the development of low-cost and high-efficiency composite photocatalysts.

## Figures and Tables

**Figure 1 nanomaterials-12-01122-f001:**
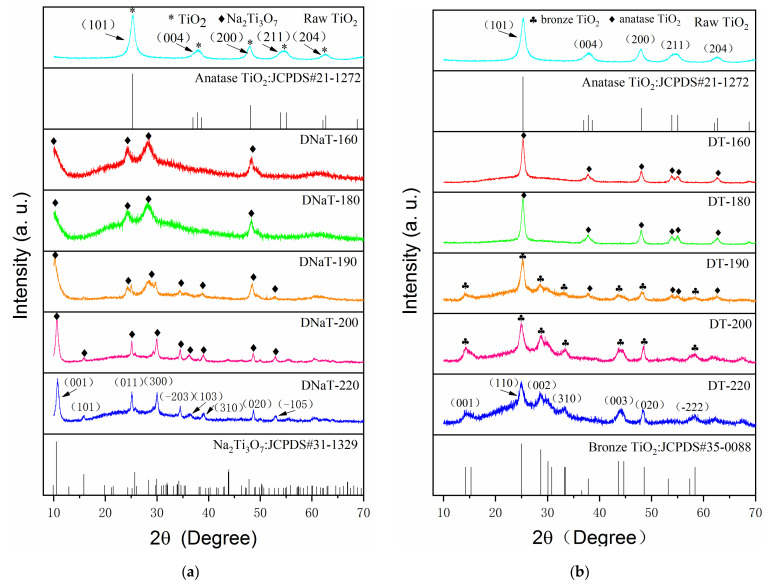
XRD diffraction patterns of the (**a**) intermediate products, * represents the diffraction peak of anatase TiO_2_, ◆ represents the diffraction peak of Na_2_Ti_3_O_7_ and (**b**) final products, ♣ represents the diffraction peak of bronze TiO_2_, ◆ represents the diffraction peak of anatase TiO_2_.

**Figure 2 nanomaterials-12-01122-f002:**
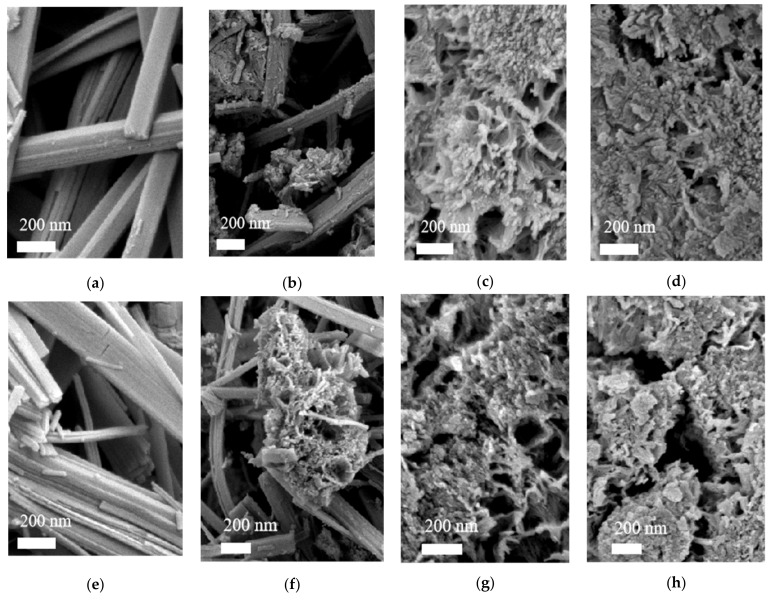
SEM, TEM, and HRTEM images of the samples: Scanning electron microscopy (SEM) images of the (**a**–**d**) intermediate products and (**e**–**h**) final products.

**Figure 3 nanomaterials-12-01122-f003:**
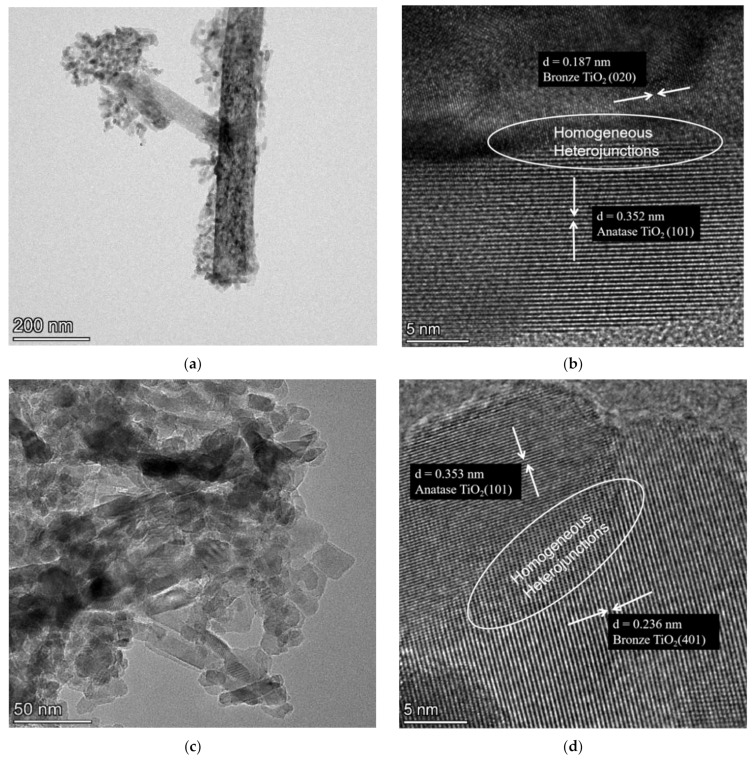
TEM images of the prepared samples: (**a**) DT-190; (**b**) High-resolution image of DT-190; (**c**) DT-180; and (**d**) high-resolution image of DT-180.

**Figure 4 nanomaterials-12-01122-f004:**
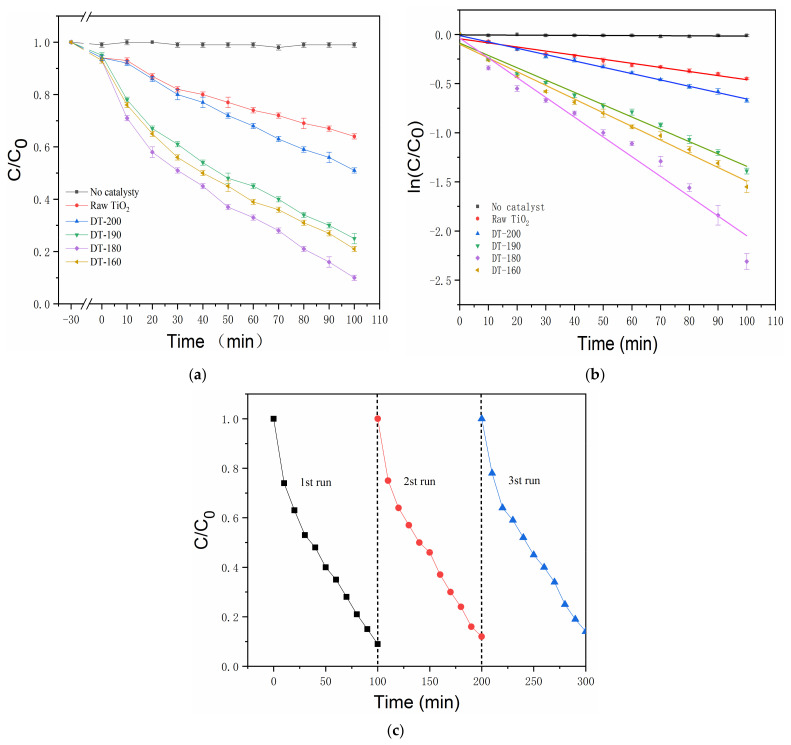
(**a**) Photocatalytic degradation curve; (**b**) Linear fittings of degrading kinetic; (**c**) Recycling performance of the DT-180 for degrading RhB.

**Figure 5 nanomaterials-12-01122-f005:**
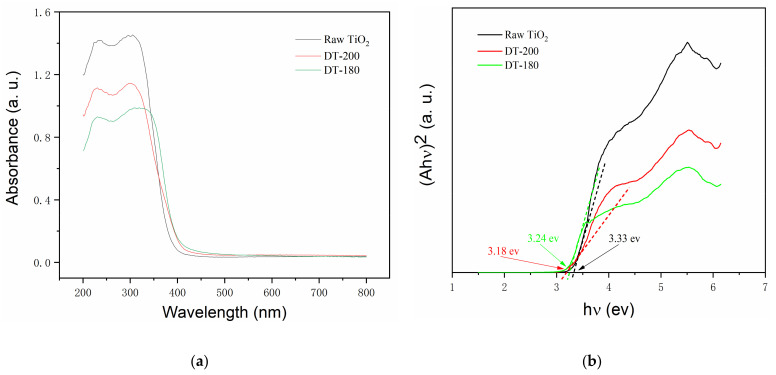
(**a**) Ultraviolet–visible (UV-vis) absorption spectra; (**b**) Tauc plots of samples to show band gap values.

**Figure 6 nanomaterials-12-01122-f006:**
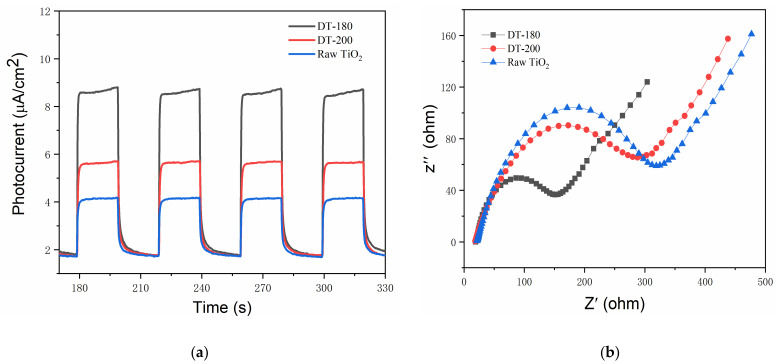
(**a**) Transient photocurrent responses. (**b**) Electrochemical impedance Nyquist plot.

## Data Availability

The data presented in this study are available on request from the corresponding author.

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
