# Peer review of "In Situ Construction of Bronze/Anatase TiO2 Homogeneous Heterojunctions and Their Photocatalytic Performances"

_nanomaterials, 2022, doi:10.3390/nano12071122_

Round 1

Reviewer 1 Report

Manuscript deals with the In-situ Construction of Bronze/Anatase TiO2 Homogeneous Heterojunctions and Their Photocatalytic performances. But publication point of view some questions to be answered.

1) There are few grammatical mistakes. Please check the manuscript for grammar and English.

2) What is novelty present work? Rewrite it at the end of introduction part.

3) Literature survey in introduction part is poor to enrich it add some recent references.

i) Journal of Colloid and Interface Science 606, (2022) 454-463, ii)  Catalysts 11 (4), (2021) 460

4) Add error bar in fig.4.

5) Add band gap plot of all prepared materials.

6) Calculate rate constant of photocatalytic reaction.

Reviewer 2 Report

In this study, the authors prepared mixed forms of TiO2 nanostructures for the potential degradation of organic dyes. In general, the article can be reconsidered for publication after a major revision only.  

1) Purity of all reagents should be disclosed, especially for those purchased from a local company. The fabrication method is not reproducible - state details, for example, what amount of raw TiO2 powder was added to the reaction!
2) I don't see adsorption experiments, i.e. samples should be mixed with RhB in the dark for ~ 30 min to reach equilibrium, and the X-axis in graph 4a should be started from - 30 min and ended at 100 min! Indicate how much dye was adsorbed on TiO2 at 0 min (i.e. exactly before the start). 
3) I don't see any statistical data, i.e. each point in Figure 4 should contain an error bar chart. Also, information on how many samples were tested per trial should be added. 
4) XRD patterns - label all diffraction peaks and calculate the mean crystallite sizes. 
5) UV-Vis data - calculate the bandgap values for different samples!  
6) Authors should test the photocatalytic activity of the samples at different pH and in the presence of interfering ions! 
7) Calculate the degradation rate for different samples. Also, show the reusability of prepared samples. 
8) Highly visible light-active materials should be discussed in the introduction. The authors can use these good examples DOI: 10.1016/j.enmm.2021.100507 and DOI: 10.3390/nano11061451 

Round 2

Reviewer 1 Report

Author tried to revise the manuscript according to reviewers comments. But still there are some correction needed before acceptance. 

1) Till there are few grammatical mistakes please correct it. example word conclusion should be conclusions.

2) Author does not write reference properly please correct it (author names wrote wrong). Reference no. 6, 22 should be 

6. Hunge, Y.M.; Uchida, A.; Tominaga, Y.; Fujii, Y.; Yadav, A.A.; Kang, S.-W.; Suzuki, N.; Shitanda, I.; Kondo, T.; Itagaki, M.; Yuasa, M.; Gosavi, S.; Fujishima, A.; Terashima, C. Visible Light-Assisted Photocatalysis Using Spherical-Shaped BiVO4 Photocatalyst. Catalysts 2021, 11, 460

22. Hunge, Y.M.; Yadav, A.A.; Kang, S.-W.; Kim, H., . Photocatalytic degradation of tetracycline antibiotics using hydrothermally synthesized 349
two-dimensional molybdenum disulfide/titanium dioxide composites. Journal of Colloid and Interface Science, 2022,606,454

Reviewer 2 Report

No more comments 

Author Response

Dear reviewer.

Thank you very much for your review. Your professional suggestions have helped me a lot. Thank you again.

Best regards,

Yong Li